# Electric Arc Furnace Dust Recycled in 7075 Aluminum Alloy Composites Fabricated by Spark Plasma Sintering (SPS)

**DOI:** 10.3390/ma15196587

**Published:** 2022-09-22

**Authors:** Elder Soares, Nadège Bouchonneau, Elizeth Alves, Kleber Alves, Oscar Araújo Filho, David Mesguich, Geoffroy Chevallier, Nouhaila Khalile, Christophe Laurent, Claude Estournès

**Affiliations:** 1Mechanical Engineering Department, Federal University of Pernambuco (UFPE), Av. da Arquitetura, s/n, Recife 50740-550, PE, Brazil; 2CIRIMAT, Université de Toulouse, CNRS, Université Paul-Sabatier, 118 Route de Narbonne, CEDEX 9, 31062 Toulouse, France; 3Plateforme Nationale CNRS de Frittage Flash (PNF2), Module de Haute Technologie, Université Toulouse 3—Paul-Sabatier, 118 Route de Narbonne, CEDEX 9, 31062 Toulouse, France

**Keywords:** electric arc furnace dust, waste recycling, powder metallurgy, spark plasma sintering, aluminum matrix composites

## Abstract

The reuse of industrial waste, such as electric arc furnace dust (EAFD) as reinforcement in aluminum matrix composites (AMC), is still little explored even though it has shown potential to improve the mechanical properties, such as hardness and mechanical strength, of AMCs. To propose a new alternative for EAFD recycling, AA7075-EAFD composites were produced by spark plasma sintering (SPS). The starting powders were prepared by high-energy milling with different weight fractions of EAFD in two particle size ranges added to an AA7075 matrix. SEM shows that the distribution of reinforcement particles in the matrix is homogeneous with no agglomeration of the particles. XRD patterns of initial powders and the SPS-sintered (SPSed) samples suggest that there was no reaction during sintering (no additional peaks were detected). The relative density of all SPSed samples exceeded 96.5%. The Vickers microhardness of the composites tended to increase with increasing EAFD content, increasing from 108 HV (AA7075 without reinforcement) up to 168 HV (56% increase). The maximum microhardness value was obtained when using 15 wt.% EAFD with a particle size smaller than 53 μm (called G1), showing that EAFD presents a promising potential to be applied as reinforcement in AA7075 matrix composites.

## 1. Introduction

Electric arc furnace (EAF) technology is used for the production of steel from recycled ferrous scrap mixed with cast iron and/or direct reduced iron, but in the process, it also produces 15–25 kg of electric arc furnace dust (EAFD) per ton of steel [1,2]. EAFD, which is considered a waste product, is a generic name for powders mainly composed of oxides (principally ZnO, Fe_3_O_4_/ZnFe_2_O_4_), and their composition may vary on a day-to-day basis depending on the process conditions including feedstock composition, furnace temperature, production time, and the type of furnace used [3]. Moreover, EAFD is classified as hazardous by several agencies, including the Brazilian Association of Technical Standards [4], the Environmental Protection Agency [5], and the European Waste Catalog [6], due to its content of metals such as zinc, cobalt, copper, lead, and cadmium. Therefore, despite it being considered a renewable resource [7] and a secondary raw material [8], finding a suitable use for the EAFD represents a strong challenge, and some studies have reported recycling it as filler in composite materials with polymer [9,10], ceramic [11], cement [12], concrete [7], and metal matrix [13,14,15]. Regarding the latter, aluminum matrix composites (AMCs) in particular are of great interest because alloys such as the 7075 aluminum alloy (AA7075) exhibit characteristics (low density, high strength-to-weight ratio) that are of great importance, including for applications in the automotive, aeronautics, and naval industries. Many compounds have been used for the reinforcement of AA7075, including SiC [16,17,18,19], TiC [17,20], B_4_C [21,22], Zr [23], Ag [24], TiO_2_ [25], Al_2_O_3_ [26,27], and reduced graphene oxide [28]. However, such compounds may themselves be of high value and therefore expensive, whereas the use of agricultural [27,29,30,31,32,33,34] and industrial waste [13,14,15,35] as reinforcement in AMCs is still little explored. The aim of this work is to investigate AA7075-EAFD composites with different EAFD contents and two different particle size ranges. Moreover, in contrast to our earlier work [15], consolidation is performed by Spark Plasma Sintering (SPS), an innovative technique that allows one to produce dense samples with fine microstructures at temperatures lower than most conventional sintering techniques in relatively short processing times [36]. The novelty of this research consists in giving a destination to EAFD, which has a considerable environmental impact because it contains elements harmful to human health, by using it as reinforcement in aluminum alloy 7075.

## 2. Materials and Methods

### 2.1. Powders

An AA7075 powder with an average particle size of 30 µm was purchased from Alcoa (Brazil). Its element composition was investigated in a previous work [37]: Al (91.48 wt.%), Zn (5.06 wt.%), Mg (1.71 wt.%), Cu (1.38 wt.%), Cr (0.27 wt.%), and Si (0.10 wt.%). The raw EAFD (supplied by the Gerdau Aço Norte plant, Recife, State of Pernambuco, Brazil) was separated by sieving with the sieve sequence used was 65, 150, 200, and 270 mesh. The batches used in the present study correspond to the two finer size ranges: batch “−270” (smaller than 53 μm, hereafter noted as G1) and batch “−200/+270” (53–75 μm, hereafter noted as G2). Starting powders were prepared with 0, 5, 10, and 15 wt.% EAFD, using either G1 or G2, mixed with the AA7075 matrix, totaling 30 g for each sample. To perform the high-energy milling, each mixture was filled in a 304 L stainless steel jar along with 2 wt.% (0.6 g) stearic acid (CH_3_(CH_2_)_16_COOH, 45 mL isopropyl alcohol, and 300 g of SAE 52100 steel balls of approximately 6.2 mm diameter (ball-to-powder weight ratio of 10:1) [27]. After this, the jar was closed and placed in a high-energy mill (SPEX). The powders were milled for 1 h at 720 RPM. Subsequently, drying was performed in an oven at 150 °C for 15 min. The nomenclature of the starting powders according to their contents is presented in Table 1.

### 2.2. Spark Plasma Sintering (SPS)

The powders were consolidated using an SPS device (Fuji 632Lx, Fuji Electronic Industrial CO., Saitama, Japan) available at the Plateforme Nationale de Frittage Flash located at the Université Toulouse 3 Paul Sabatier. The powders were loaded into a graphite die and a graphitic foil was placed at the punch/powder and matrix/powder interfaces to allow for easy removal after sintering. The SPS cycle was carried out in a vacuum (residual cell pressure < 5 Pa) utilizing a 40 ms: 7 ms direct current pulse pattern (pulse on: pulse off). The temperature was controlled using a K-thermocouple presented in an opening (3 mm deep) penetrated on the external surface of the die. The samples were heated (100 °C/min) up to 550 °C, and this temperature was maintained for 30 min. A uniaxial pressure of 100 MPa was applied at room temperature and kept constant during the heating and dwell steps. Natural cooling was applied until room temperature was reached and the uniaxial load was progressively relaxed. SPS pellets had a diameter of 20 mm and a thickness of 3 mm.

### 2.3. Characterization

The element composition of the EAFD powders (G1 and G2) was determined by X-ray fluorescence (XRF) spectrometry (XRF, BRUKER S2 Ranger, Karlsruhe, Germany). The density of the powders was evaluated by He pycnometry (Micromeritics AccuPyc II 1340, Norcross, USA). Ten He purges were performed for each sample in order to obtain a stable value. The particle size analysis of G1 and G2 was performed in a liquid medium (water) by laser diffraction (Mastersize 2000, Malvern Instruments, Malvern, UK). The crystallized phases were detected and identified using X-ray diffraction (XRD, Bruker D4, Karlsruhe, Germany, in θ-2θ configuration), using CuK_α1_ radiation (0.15406 nm). The density of the SPSed samples was determined using Archimedes’ principle and a hydrostatic balance (Sartorius MSE224S-YDK03, Göttingen, Germany) with 10 measurements taken for each sample. Densifications were calculated using 2.81 g/cm^3^ for AA7075 [38]. Optical microscopy (Keyence VHX-1000 Digital Optical Microscope, Osaka, Japan) and field-effect-gun scanning electron microscopy were used to examine the materials (FESEM, JEOL 6700F, Tokyo, Japan). For microstructural observations, grinding was done with SiC papers (grades 1200 and 2400). The samples were then polished in four stages using diamond suspensions of 6, 3, and 1 μm and an amorphous silica suspension of 20 nm. Etching (10 g NaOH + 100 mL H_2_O for 10 s) was done at room temperature to show the grain boundaries. The samples were soaked in ethanol for 20 s before being immersed in distilled water for 1 minute with ultrasonic agitation to cease the etching process. Vickers microhardness tests were performed (Mitutoyo HM200, Kawasaki, Japan) with a load of 0.1 kg applied for 10 s.

## 3. Results and Discussion

The examination of the element composition of the G1 and G2 powders (Table 2) reveals some differences between the two samples, the main one being that G1 contains less iron than G2 (39.89 vs. 45.33 wt.%, respectively) and more zinc than G2 (36.93 vs. 28.80 wt.%, respectively).

The density found for G1 (3.7522 ± 0.0007) is significantly higher than that found for G2 (3.411 ± 0.003), which is likely due to their different composition (as discussed below in the X-ray diffraction analysis). Therefore, 5, 10, and 15 wt.% EAFD powder in the sample correspond to 3.8, 7.7, and 11.7 vol.% for G1 and 4.2, 8.4, and 12.7 vol.% for G2.

The particle size distributions (PSD) for G1 and G2 are shown in Figure 1. For G1 (Figure 1a), there is a bimodal distribution with fines formed by particles in the size range of 0.2–13 μm and larger particles in the range of 13–120 μm, each corresponding to approximately 50% of the volume. The average particle size is equal to 14 μm. For G2 (Figure 1b), the proportion of fines (0.2–17 µm) is much less pronounced than for G1, representing only around 12% of the accumulated volume. The larger particles are in the range of 17–158 μm, and the average size is equal to 63 μm.

Analysis of the XRD patterns (Figure 2) reveals the presence of ZnO and Fe_3_O_4_ and/or ZnFe_2_O_4_ as major compounds for both G1 and G2. Fe_3_O_4_ and ZnFe_2_O_4_ share the spinel structure with close cell parameters, and it cannot be concluded here if it is one or the other compound. Using Mössbauer spectroscopy, Machado and collaborators [39] were able to verify the presence of these two phases in an EAFD sample with a finer particle size (average diameter 1.88 μm), the Fe_3_O_4_ and ZnFe_2_O_4_ representing 14% and 29% of the EAFD, respectively. There is also more free SiO_2_ in G2 than in G1, which is in agreement with the silicon content found by XRF (Table 1).

As presented earlier, the difference in density between G1 and G2 was attributed to their compositions. Indeed, as shown in Figure 2, EAFD powders are mainly composed of ZnFe_2_O_4_ and/or Fe_3_O_4_, ZnO, and SiO_2_. Table 2 shows that G1 has a higher concentration in Zn compared to G2, which is significant because ZnO and ZnFe_2_O_4_ have the highest density among the oxides constituting the EAFD powders used, with densities of 5.61 g/cm^3^ and 5.34 g/cm^3^, respectively. G1 also has a lower concentration in Fe and Si compared to G2, and the corresponding oxides Fe_3_O_4_ and SiO_2_ have lower densities of 5.17 g/cm^3^ and 2.65 g/cm^3^, respectively. Therefore, G1, compared to G2, has higher amounts of denser oxides (and lower amounts of lighter oxides), leading to higher density for G1.

FESEM images (Figure 3) of G1 and G2 reveal that they are made up of spherical agglomerates about 50–80 µm in diameter, with submicronic particles in the 0.1–1 µm range. This suggests that the agglomerates could be relatively easily broken by milling. Few rod-like particles are also observed (about 300 µm long and 30 µm wide), notably for G2 (inset in Figure 3c). Several authors have verified the presence of these spherical agglomerates formed by fine particles or fine particles covering larger particles. Rocabois has pointed out spherical shapes formed from spinel-type metal oxides XFe_2_O_4_ (X = Fe, Zn, or Mn) [9,15,40].

FESEM images (Figure 4) of the AA7075 powder milled during 1 h show that it is made up of micrometric flakes (lateral dimensions 10–100 µm, thickness about 1 µm), which is a typical morphology of milled ductile metals.

For the composite powders, the EAFD particles are homogeneously dispersed at the surface of the AA7075 flakes (Figure 5). No micrometric EAFD agglomerates are observed. Figure 5a and Figure 5b show, respectively, the appearance of the starting powders 15G1 and 05G2 at low magnification. Figure 5c,d show higher magnification of selected areas in Figure 5a and Figure 5b, respectively. It can be seen that these powders are very similar, differing only in the surface area covered by EAFD on the AA7075 flakes.

All the starting powders have been consolidated by SPS, and the XRD patterns of the starting powder 15G1 and sintered samples 15G1-SPSed and AA7075-SPSed are reported in Figure 6. For the starting powder 15G1, Figure 6a shows intense Al peaks and weak peaks attributed to the various compounds initially present in the G1/EAFD powder (Figure 2a), even for this sample with 15 wt. % G1. The similarity between the XRD patterns of 15G1 starting powders (Figure 6a) and the 15G1-SPSed sample (Figure 6b) suggests that no reaction took place during sintering as no new phase was detected. The XRD patterns of the sintered G2/EAFD—AA7075 composites have similar features (not shown). The XRD pattern of the AA7075-SPSed is shown in Figure 6c.

In view of the fact that no reaction between the compounds is detected on the XRD (i.e., no additional peaks, thus no new phase present), the theoretical density of the samples can be calculated via a law of mixtures. All the consolidated samples by SPS exhibit densification higher than 96.5% (Table 3).

Optical images of the composites (Figure 7) reveal the compacted aluminum flakes with a size similar to those observed for the AA7075 powder. It is not possible to observe the EAFD particles at this magnification.

Figure 8 shows the FESEM images of AA7075 fractured samples (Figure 8a,b), 15G1 (Figure 8c), and 15G2 (Figure 8d). In these fractured samples, we find lamellar morphology with locally aligned lamellae (clearly visible in Figure 8a,c). The cohesion between AA7075 grains seems to be high (low presence of porosity), which is in agreement with relative densities of about 97–100%, the residual porosity probably being interlamellar. Factors such as processing conditions, morphology, and reinforcement distribution influence the fracture modes of composites with single or multiple reinforcements. The porosity and roughness of the sample also have an effect on the fracture mode. For all samples, the fracture surfaces indicated the presence of the mixed transgranular-intergranular fracture mode. The microcracks indicated by the yellow arrows in Figure 8c,d (15G1-S and 15G2-S, respectively) indicate that the failure is initiated between the lamellar particles of the matrix due to microvoids and/or the presence of brittle precipitates at these interfaces since the reinforcement is formed by oxides [41].

The Vickers microhardness (Figure 9) increases considerably upon the increase in EAFD content, from about 110 HV to 170 HV (with G1) and 150 HV (with G2), thus 55% and 40% higher than AA7075, respectively. The strengthening mechanism is possibly oxide dispersion strengthening (ODS). The high concentration of Fe and Zn enables the formation of high-hardness oxides. The dispersion of oxides in the matrix contributes to impeding the movement of dislocations, thus increasing the strength and hardness properties. In previous work, Alves et al. [15] strengthened the AA7075 alloy by adding EAFD using a conventional powder metallurgy technique of cold uniaxial pressing and sintering in a furnace subjected to a nitrogen atmosphere. Under established experimental circumstances, the inclusion of 5% EAFD resulted in a considerable increase in Vicker microhardness (HV), from 85.09 to 124.6, compared to the unreinforced alloy. The addition of waste to the aluminum matrix increased the average Young’s modulus (GPa) values. The Young’s modulus (E) values found for the AA7075 alloy and the AA7075-5%EAFD composite were 35.51 and 70.48 GPa, respectively.

These results demonstrate that both EAFD size ranges (G1 and G2) as reinforcement in aluminum matrix composites can be another alternative to recycling this residue from the steel industry. In the present work, an increase in microhardness (Hv) up to 15 wt.% of EAFD was observed. In the AA7075-S SPSed sample, the Vickers microhardness value obtained was 108 Hv, whereas, in composites with 15% EAFD, the microhardness reached 168 Hv using G1 (15G1-S, EAFD with particle sizes smaller than 53 µm) and 152 Hv using G2 (15G2-S, EAFD with particle sizes between 53 µm and 75 µm). Studies in the literature that deal with the use of EAFD as reinforcement in aluminum matrix, it is observed that the improvement of mechanical properties as a function of the content of this residue only occurs up to the limit of 10 wt.% of EAFD [13,14]. This difference may be related to the technique of Spark Plasma Sintering used in this research, which has several advantages over conventional production techniques [36,42], among them the ability to produce materials with very low porosity and to limit grain growth. Using conventional uniaxial compaction and sintering powder metallurgy techniques, Flores-Vélez et al. [13] reported that, when using EAFD as reinforcement in an Al matrix, the hardness of the composite increases as a function of EAFD content when using up to 10 wt.% of this reinforcement. The hardness of the unreinforced matrix is approximately 52 HVN while when 15 wt.% EAFD was used as reinforcement, the value rose to about 74 HVN. When using a larger amount of this reinforcement (20%), the microhardness was close to the value of the unreinforced matrix. This poor performance of the reinforcement above 10 wt.% may be associated with the low relative density obtained in this work, which did not exceed 85%. According to these authors, this low relative density at the end of the process is associated with the size of the EAFD particles, which are in the nanometric range, inhibiting higher densification, i.e., the larger the amount of EAFD, the lower the relative density obtained. In contrast, the relative density of the EAFD/AA7075 composites produced by us using the SPS technique ranged from 96.5% to 100% with a much shorter production time, showing the superiority of this technique over conventional powder metallurgy techniques.

The possible application of the AA7075 matrix composites reinforced with G1 and G2 is in parts requiring low density (the highest density obtained was 2.89 g/cm^3^ using 15 wt.% of G2, which is only slightly higher than the alloy density) and improved wear resistance. Possible applications for this material are as follows: small components in near net shape for the aerospace and automotive industries (such as pistons for automobile suspension shock absorbers).

## 4. Conclusions

EAFD-AA7075 composites with different EAFD contents (0, 5, 10, and 15 wt.%) and two particle sizes (noted G1 and G2) were sintered by SPS (550 °C, 100 MPa, 30 min). The samples produced by SPS were almost or fully dense, and XRD analysis showed that no reaction took place between EAFD and the AA7075 matrix during sintering. The morphology of powders is retained in the bulk samples, with EAFD particles dispersed among AA7075 micrometric grains. Low porosity was observed along the AA7075 lamellae that exhibit good cohesion. For both EAFD size ranges, the Vickers microhardness considerably increases upon the increase in EAFD content (using 15 wt.% of G1 or G2, hardness is 55% and 40% higher than for AA7075, respectively), showing that using EAFD powder as reinforcement in aluminum matrix composites could be an interesting path and a promising recycling alternative for this residue.

The use of EAFD as reinforcement in AA7075 matrix composites seems to produce improvements in the mechanical properties in relation to the unreinforced aluminum matrix, but it is necessary to carry out more detailed studies to prove this trend. In further studies, it would be interesting to analyze the effect of the milling time of the starting powders and other SPS routes, verifying their influence on the microstructure, mechanical properties, and relative density of EAFD/AA7075 composites. In addition, it would also be important to investigate the effect of some heat treatments on the hardness and mechanical strength of the final product.

## Figures and Tables

**Figure 1 materials-15-06587-f001:**
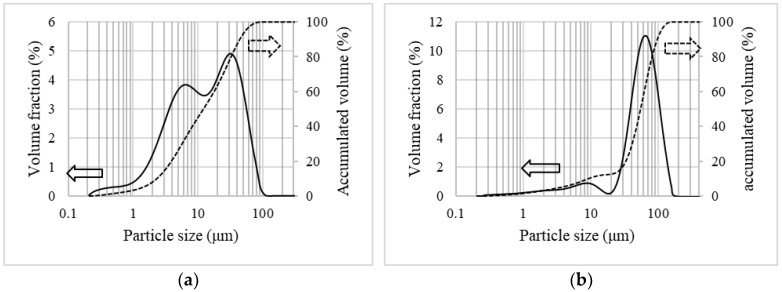
Particle size distribution for the EAFD powders: (**a**) G1 [15] and (**b**) G2. The solid lines represent the particle size distribution, and the dashed lines indicate the accumulated volume.

**Figure 2 materials-15-06587-f002:**
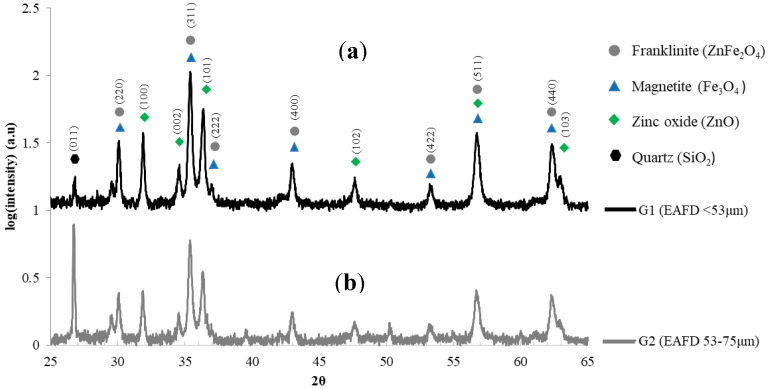
XRD patterns of the EAFD powders: (**a**) G1 and (**b**) G2.

**Figure 3 materials-15-06587-f003:**
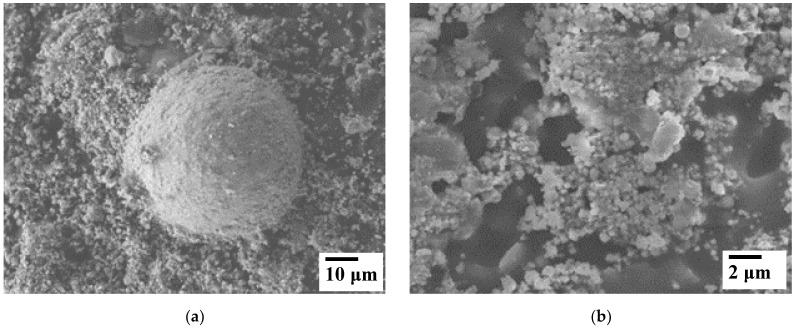
FESEM images of the G1 (**a**,**b**) and G2 (**c**,**d**) EAFD powders. The inset in (**c**) shows a rod-like particle observed.

**Figure 4 materials-15-06587-f004:**
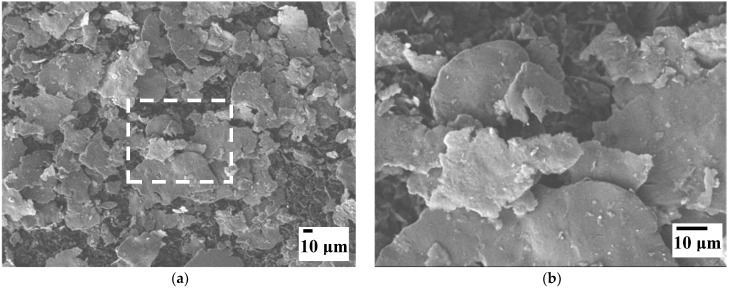
FESEM images of the AA7075 powder. (**b**) is a magnification from the boxed area in (**a**).

**Figure 5 materials-15-06587-f005:**
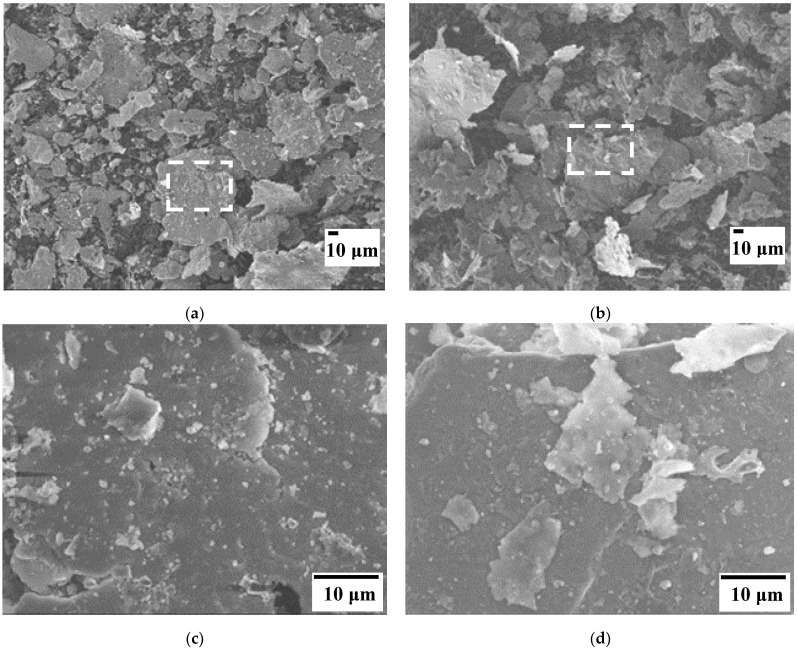
FESEM images of selected starting powders: (**a**) 15G1; (**c**) is a magnification from the boxed area in (**a**); (**c**) 05G2; (**d**) is a magnification from the boxed area in (**b**).

**Figure 6 materials-15-06587-f006:**
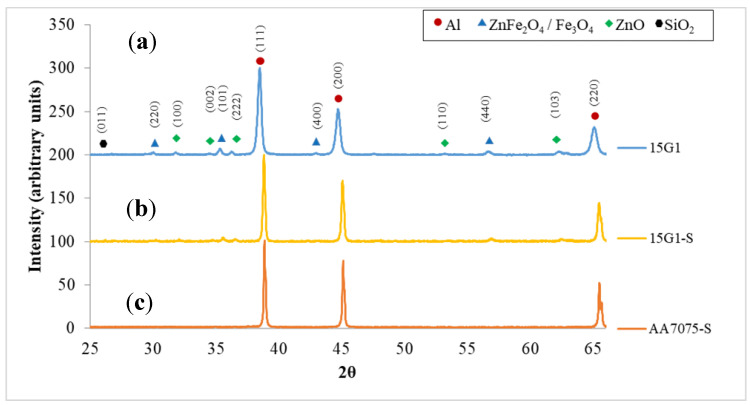
XRD patterns of the: (**a**) 15G1 starting powder; (**b**) 15G1-S; and (**c**) AA7075-S SPSed samples.

**Figure 7 materials-15-06587-f007:**
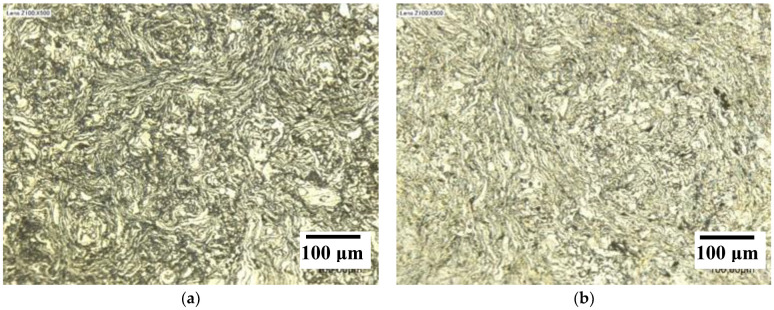
Optical image of selected EAFD/AA7075 composites: (**a**) 10G1-S; (**b**) 10G2-S.

**Figure 8 materials-15-06587-f008:**
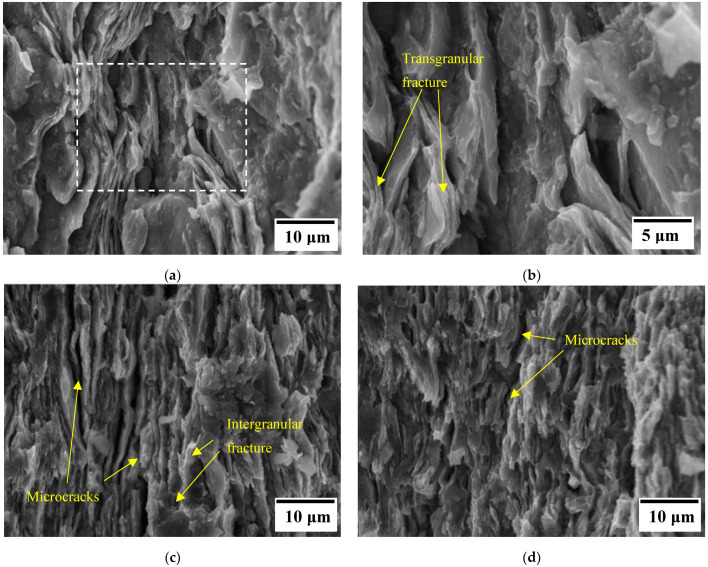
Fracture surface FESEM images of sintered samples: (**a**) AA7075; (**b**) boxed area in (**a**); (**c**) 15G1-S; (**d**) 15G2-S.

**Figure 9 materials-15-06587-f009:**
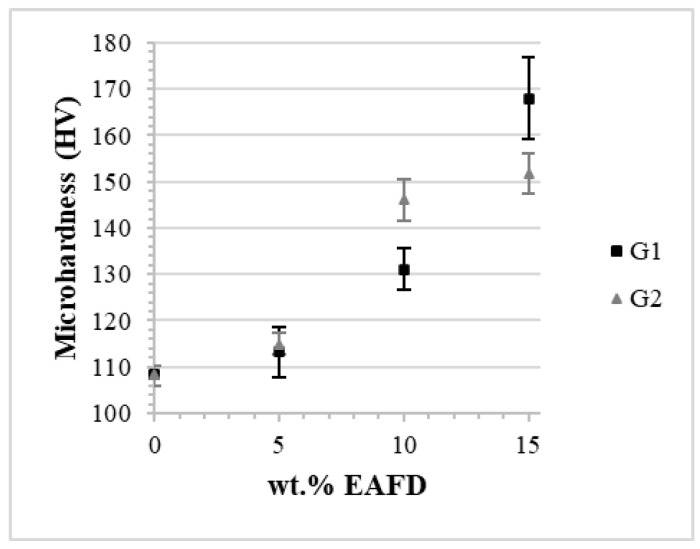
Vickers microhardness of G1, G2, and AA7075 samples vs. EAFD content.

**Table 1 materials-15-06587-t001:** Nomenclature of the starting powders according to their compositions.

Sample	Content (wt.%)		
	G1	G2	AA7075
AA7075	-	-	100
05G1	5	-	95
10G1	10	-	90
15G1	15	-	85
05G2	-	5	95
10G2	-	10	90
15G2	-	15	85

**Table 2 materials-15-06587-t002:** Composition of the EAFD powders from X-ray fluorescence (wt.%).

	Fe	Zn	Ca	Mn	Si	Mg	Al	Pb	Gd	La
G1	39.89	36.93	7.00	4.13	2.74	2.69	1.48	1.22	0.66	0.65
G2	45.33	28.80	7.66	2.99	4.98	2.90	2.02	1.22	0.40	-
	**Cr**	**K**	**Cu**	**Cl**	**Ti**	**Ba**	**S**	**P**	**Others**	
G1	0.52	0.43	0.31	0.27	0.19	0.17	0.16	0.12	0.44	
G2	0.40	0.52	0.30	0.24	0.69	0.29	0.28	0.25	0.73	

**Table 3 materials-15-06587-t003:** Calculated (*ρ_c_*) and experimental relative density (*ρ*) of the 20 mm sintered samples.

Sample	G1 (wt.%)	G2 (wt.%)	AA7075 (wt.%)	*ρ_c_* (g∙cm^−3^)	*ρ* (%)
AA7075-S	-	-	100	2.81	99.3 ± 0.1
05G1-S	5	-	95	2.85	96.5 ± 0.2
10G1-S	10	-	90	2.88	97.5 ± 0.1
15G1-S	15	-	85	2.92	98.2 ± 0.1
05G2-S	-	5	95	2.83	99.1 ± 0.1
10G2-S	-	10	90	2.86	97.2 ± 0.1
15G2-S	-	15	85	2.89	100.0 ± 0.3

## Data Availability

Data sharing not applicable.

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
