# Peer review of "Electric Arc Furnace Dust Recycled in 7075 Aluminum Alloy Composites Fabricated by Spark Plasma Sintering (SPS)"

_materials, 2022, doi:10.3390/ma15196587_

Round 1

Reviewer 1 Report

In the current study, the authors have focused on ground-breaking research but found technical data to be lacking. However, it should be included. From the manuscript, it was noted that the author discussed microstructure, FESM, and XRD analysis, but relevant technical terms are missing. Furthermore, it was observed that only one property (hardness) was discussed in the manuscript. It is not accepted . The author needs to incorporate more properties to enhance the quality of the research paper and hence it may be recommended for further publications after incorporating all. The following comments are given below.

1.In the abstract, it is inferred that hardness and strength and it is generally written specifying whether it's physical or mechanical strength. Also, it was specified as a different fraction to justify whether it is a weight fraction. or volume fraction.

2. Abstract needs to be rewritten and it must be optimized

3. In the introduction, it was specified as ray material. Is its raw material or ray material? Check?

4. The novelty of the current research should be strengthened and the applications for which it was developed should be described in detail

5. In material and methods, After ball milling what is the particle size of matrix powder as well as reinforcement used also infer the particle size image in the manuscript for both in order to enhance the quality of the research paper .

6. Experimental methods from raw material to sintered composites – processing methods need to be elaborated in detail.  A suitable Citation must be added for this section.

7. It was pointed out in the manuscript that the authors described in more detail the equipment used to analyze the mechanical properties of the synthesized composites, but the results of the experiments are scaled back in the manuscript. Why?

8. In the Result and discussion G1 and G2 chemical composition has been inferred –Justify it

9. The density found for G1 (3.7522 ± 0.0007) is significantly higher than that found for 124 G2 (3.411 ± 0.003), which is likely due to their different composition- How?

10. Incorporate the density of synthesized composites using G1 and G2 and discuss in detail how they can be used for functional applications.

11. XRD discussion in the text does not match the actual XRD pattern, and the estimate of crystallite size is missing.

12. On page no 6/12 the authors have stated Error! Reference source not found- many times – this will make the readers and reviewer go away from the subject. The author should clearly state the technical terms and the same needs to be incorporated into the manuscript

13. The authors have discussed the strengthening the mechanism of synthesized composites- It is mandatory to discuss.

14. The research paper needs to be rephrased in the sequence of order a. Material and methods ( Proper justification needs to give about elements and size of powders)- Microstructure-SEM-XRDA-Mechanical properties- but in the manuscript it was randomly cooked.

15. No need for more stories about the enhancement of Vickers hardness. What the reason to increase such hardness should be incorporated.

16. Conclusion needs to optimize

17.  From the manuscript it was observed that only one property (hardness) was discussed in the manuscript. It is not accepted. The author needs to incorporate more properties to enhance the quality of the research paper. 

Author Response

Please find attached the reponse to the reveiwer 1 comments

Reviewer 2 Report

The article on electric arc furnace dust  recycled in 705 aluminum alloy composites by spark plasma sintering shown potential improvement compare to other fabrication methods.

Author Response

The authors would like to thank the reviewer for his report and publication recommendation.

Reviewer 3 Report

Good paper. The recycling of EAFD is an important problem to solve. The paper is written clearly, in a logical way. Minor English corrections needed.

Technical remarks:

11. Mass of the milled powders is not given (BPR is not sufficient for repetition of the experiment).

22. Page 4: “silicium content”, should be : silicon

33. The chemical composition determined by XRF does not take into account the oxygen content (there are oxides and spinel-type phases rich in oxygen in the structure of EAFD). Therefore, do the numbers  presented in Table 2 results from re-calculation procedure.

   Merit question:

11. What is a reason of differences in Fe/Zn content in G1 and G2 powders? Is it influenced by the differences in particle size only?

Author Response

Please find attached the response to reviewer 3 comments

Round 2

Reviewer 1 Report

In the present investigation, the author focused on electric arc furnace dust recycling 7075 aluminium composites. The novelty of the work was included and the correction given was incorporated into the manuscript. However, it was noted that at least 5 mechanical properties of synthesized composites are expected for characterization study but it was not observed in the manuscript instead it was focused on the detail of morphology od synthesized composites. In future, further characterization needs to perform and same needs to be included in the manuscript.